**Subject Category:**
Biology (whole organism)

behaviour/chemical biology/cognition

sexual communication, lipid component, *Euscepes postfasciatus*, survival analysis, same-sex sexual behaviour

**Author for correspondence:**
Haruki Tatsuta
e-mail: htatsuta@agr.u-ryukyu.ac.jp

†Present address: Faculty of Medicine, University of the Ryukyus, Uehara, Okinawa 9030213, Japan.

# When a male perceives a female: the effect of waxy components on the body surface on decision-making in the invasive pest weevil

## Mutsumi Isa[1,†], Norikuni Kumano[2] and Haruki Tatsuta[1,3]

[1]Department of Ecology and Environmental Sciences, Faculty of Agriculture, University of the Ryukyus, Nishihara, Okinawa 9030213, Japan
[2]Laboratory of Insect Ecology, Obihiro University of Agriculture and Veterinary Medicine, Obihiro, Hokkaido 0808555, Japan
[3]The United Graduate School of Agricultural Sciences, Kagoshima University, Korimoto, Kagoshima 890-8580, Japan

NK, 0000-0003-2803-8660; HT, 0000-0001-8847-8874

Insects use various semiochemicals for sexual communication and mate recognition; these can therefore be used to govern the behaviours of harmful pest species, and several candidate chemicals have been explored for this purpose. For the West Indian sweet potato weevil, *Euscepes postfasciatus*, which is one of the most serious pests of sweet potato, no effective capture techniques, such as sex pheromone lures, exist. Toward exploring promising procedures for monitoring these weevils, we assessed the effect of secretions on the body surface on the recognition of congeners and on courtship behaviour in the weevils. Our study clearly demonstrated that weevils responded to extracts from the body surface, and the behaviour adopted by the weevils varied significantly depending on the condition of the extracts. Furthermore, we found a significantly prolonged retention time for males on glass beads covered with extracts of females based on survival analysis. These findings are, as far as we are aware, the first to show the effect of lipid components of the body surface on decision-making in these economically important pest weevils.

## 1. Introduction

Considerable ecological, behavioural and physiological information regarding the hydrocarbons on the body surfaces of insects has recently been accumulated (e.g. [1]). Hydrocarbons not only are embedded in the cuticular lipid layers of all insects

[2] and function to prevent water loss and desiccation [3] and infection by microorganisms [1], but also are used for mutual communication in individuals belonging to different sexes in a great variety of insects [4,5]. In particular, they are known to be semiochemicals that play various crucial roles, such as providing information regarding the socio-sexual status of males, as well as species and gender recognition [1,6,7]. Hydrocarbons are commonly involved in mediating close-range mate recognition (e.g. [8,9]) because they convey critical cues for fertility and sex (and more) in various insects [10]. From the view of practical use in pest management, such semiochemicals, as well as volatile sex pheromones, can be used for the orientation and trapping of harmful insect pests (e.g. [11,12]).

The West Indian sweet potato weevil, *Euscepes postfasciatus* (Fairmaire) (Coleoptera: Curculionidae), is one of the major pests of the sweet potato, *Ipomoea batatas* (L.) Lam. in the South Pacific, the Caribbean basin and some countries of Central and South America [13]. The first record of the invasion of *E. postfasciatus* into Japan was made in the Ogasawara Islands in 1905 [14], while the first detection in the Southwestern Islands was recorded in 1947 in the middle regions of Okinawajima Island [15]; the species is currently already distributed in the Amami and Ryukyu Islands [16]. To prevent further expansion of the species distribution into other areas of Japan, the transportation of the pest plants from affected regions is strictly prohibited by the Japanese Plant Protection Law. While a sterile insect technique (SIT) programme for the pursuit of eradication of this weevil is currently in progress in Okinawa Prefecture [17], no promising chemicals mimicking sex pheromones that can be used to attract conspecific individuals have yet been identified. This weevil is known to be weakly attracted to light-emitting diodes or chemiluminescent light (chemiluminescent: [18], ultraviolet: [19], green and other colours: [20]); however, the efficacy of capture is still insufficient for monitoring the weevil; thus, there is no firm protocol to evaluate successful eradication using the SIT programme.

Based upon observation so far, the sequence of mating behaviour in *E. postfasciatus* males is characterized by pre- and post-copulation guarding behaviour [21–23]. Although the causal relationship between the pre- and post-copulation guarding behaviours and the role of waxy components on the body surface during mating has never been clarified in this weevil, mating behaviours and the relevant semiochemicals, such as hydrocarbons, might exert an influence on mating success and reproductive precedence, as has been revealed in other insects (e.g. [24–28]). In the present study, we aimed to assess the possible functions of the waxy compounds covering the whole body of the insect, possibly including several types of hydrocarbons, in terms of conspecific interaction and gender identification. After defining the sequence of mating behaviour, we extracted waxy compounds and performed behavioural assays using these extracts. We also conducted a mate-choice experiment using live adults (two males and a female). Based on these experiments, we attempt to conjecture the function of waxy compounds from the body surfaces of weevils in the recognition of congeners and the timing of awareness of the partner's gender. We should also note that a high frequency of same-sex (male–male in this case) sexual behaviour (SSB) [29,30], which has not been paid attention to in this weevil so far, was observed. Therefore, we reported both male–male and male–female sexual behaviour in the present analysis. The present study would considerably aid the identification of effective attractants for the weevils and thus for the establishment of successful pest management.

# 2. Material and methods

## 2.1. Insects

We used a strain of *E. postfasciatus* originally captured at Yomitan village (26°24′ N, 127°43′ E) in May 2006 and maintained at $25 \pm 1°C$ under a photoperiod of 14 L : 10 D (lights on from 04.00 to 18.00 h) at a facility of the Okinawa Prefectural Plant Protection Center (OPPPC), Japan. Weevils were randomly selected from rearing containers at OPPPC in May 2012 to establish new experimental lines in the laboratory at the University of the Ryukyus (temperature: $25 \pm 1°C$, photoperiod: 13 L : 11 D, lights on from 02.00 to 15.00 h until 17 September 2013; from 00.00 to 13.00 from 18 September 2013) using plastic containers (450 ml). We supplied sweet potato roots as a food and culture medium for oviposition and growing larvae. Sweet potato roots were dissected approximately five weeks after inoculation with weevils. Because the emerged adults remain in the pupal chambers for one to two weeks for sexual maturation [31,32], we scrutinized the chambers after dissecting the sweet potato roots; newly emerged adults were removed and kept individually in Nunc™ Microwell™ 96 plates (Thermo Fisher Scientific, Waltham, MA, USA). Adults were sexed under a stereomicroscope within

**Figure 1.** Three sequential stages of mating behaviour in *Euscepes postfasciatus*. Each behavioural stage was defined according to Kumano *et al.* [22,23] and preliminary observation. Broken arrows indicate an irregular order of behaviours that were observed in the present study.

24 h of removal, according to the method of Kohama and Sugiyama [33], and the males and females were then preserved separately in plastic cups (84 ml) with a piece of sweet potato root (approx. 3 cm diameter, 1 cm thickness) to prevent interference and casual mating. Newly collected weevils were designated a substantial age of 0-day-old [34]. We used sexually matured 12- to 17-day-old virgin weevils for subsequent behavioural studies. All individuals were used only once in the experiments.

## 2.2. Definition of the stages of mating behaviours

According to Kumano *et al.* [22,35] and our observations on courtship behaviours of adult males, we have defined the pre-copulatory guarding process as the following three sequential stages summarized in figure 1. At the first stage (Stage I), after encountering each other, the male mounts the back of another individual; we defined this as the 'initial contact behaviour'. After the initial contact, the male grasps the other individual using forelegs and attempts orientation. In the experiment using beads (described in the subsequent section), we observed that weevils remained clinging to the beads after mounting and sometimes showed gestures indicating searching. We categorized these behaviours as Stage II, defined as 'post-mounting behaviour'. In addition, some weevils climbed on and off the beads several times during the observation; we simply consolidated these data as presence (i.e. mounting was observed at least once) or absence (i.e. mounting was never observed) for each experimental trial and did not take the total frequency of the behaviour into consideration. We note here that we found the female also took actions very similar to those taken by males categorized in Stage II in the present experiment (also table 1). The third stage (Stage III), defined as 'pre-copulatory behaviour', comprises different behavioural repertoires, such as rubbing the mid- and hindlegs against the posterior tip of the female's abdomen, raising the forelegs, and an attempt to insert the genitalia into the genital opening of the female. The order of the behavioural sequences from Stage II to Stage III is sometimes ambiguous; as far as we observed, males often displayed behaviours categorized as Stage III without those categorized as Stage II, and occasionally displayed behaviours categorized as Stage II after those categorized as Stage III. The establishment of copulation is finally achieved when male elevates female's abdomen, and finally the male's body is perpendicular to that of the female after the male inserts his genitalia [22,35,36]. We should note that the sum of Stage II and III may be termed as 'courtship behaviour' as frequently found in other insect species (e.g. [37]), but we occasionally found that males mounted other males (see subsequent sections), which is cognoscible as SSB [29,30]; hence, we distinguished male–male (homosexual) and male–female (heterosexual) sexual behaviour in the subsequent analysis. Here, we call male–male sexual behaviour as SSB, according to the usage of Bailey and Zuk [29]. We also recorded the exact frequency of behaviour displayed by males, but did not consider the frequency in subsequent analyses and instead converted the data into presence or absence data as in Stage I. However, we took the frequency into account when we estimated the average time of pre-copulatory behaviour for each experimental trial.

## 2.3. Behavioural experiment using live weevils

The mating behaviour was observed in a dark room kept at $26 \pm 2^\circ$C at the laboratory of the University of the Ryukyus and was recorded by means of visual judgement. Prior to the observation, randomly chosen sets of adults consisting of one female and two males of *E. postfasciatus* per trial were placed in the room for 1 h for habituation. Weevils were placed in a glass Petri dish (41.5 mm diameter and 18 mm height) for use as an arena for the observation of behaviour so as to maintain approximately the same distance from each other. The dorsal side of the male elytron was preliminarily marked using a permanent marker (PX-21, Silver, Mitsubishi Pencil, Tokyo, Japan) for sex identification [38]. At the time of behavioural observation, we used an electric torch (50 lux, light meter LX1010B; Zhangzhou WeiHua Electronic, Zhangzhou, China) covered with a red plastic film so as to minimize disturbance to the weevils. The inner side of the glass Petri dish was covered with polytetrafluoroethylene (Fulon® PTFE: Asahi Glass, Tokyo, Japan) to prevent escape of the weevils. We recorded the behaviour of individuals for 120 min in total; the behavioural observation started at 18.00 under dark conditions. Mating behaviour was recorded by visual observation. Behavioural observations were initiated immediately after placing the weevils in the dish. The riding behaviour of the female on the male and short-term contact (i.e. the partner's body was touched, but mounting was not observed) was not considered in the present analysis; however, we took SSB into account. As soon as any individual started to mount another, we removed a lone individual gently using tweezers and continued observation. We excluded pairs in which the male was inactive and did not mount on others from subsequent analysis. We recorded the presence/absence of each repertoire of post-mounting and pre-copulatory behaviour, and also the initiation and termination time of pre-copulatory behaviour. Observation was terminated either when mounting was not observed or when the duration of courtship behaviour reached 2000 s. We continued observation when pairs were still in pre-copulatory mounting when 120 min had passed from the beginning of observation. In total, data from 73 trials, composed of 32 male–male and 41 male–female pairs, were obtained.

## 2.4. Behavioural assay using hexane extracts

We conducted the experiment from 24 November 2012 to 10 November 2014 using adult weevils of experimental lines established in the laboratory at our university. Sexually matured 12–17-day-old virgin weevils were used for the extraction of cuticular lipids. To extract and determine the composition of cuticular lipids, males and females of *E. postfasciatus* were soaked in HPLC-grade *n*-hexane (1 ml; ≥98% purity; Sigma-Aldrich, St. Louis, USA) for 5 min to obtain non-polar extracts. These extracts were transferred to a clean glass vial and evaporated to approximately 30 µl under a nitrogen stream to attain a final extract concentration of approximately 2 individual equivalents µl$^{-1}$. A glass bead (3.5 mm diameter, BZ-4, AS-ONE, Tokyo) whose surface was made rough using hydrofluoric acid (46% in water; Morita Chemical Industries, Osaka, Japan) was washed with *n*-hexane (Sigma-Aldrich, St. Louise, USA) and set on the centre of a glass Petri dish (27 mm diameter and 15 mm height) covered with polytetrafluoroethylene. The size of the beads was chosen so as to simulate the size of weevils. The bead was then coated with 2 µl (approx. 4 individual equivalents) of the extracts from the cuticular surface. The amount of coated product was determined in consideration of a preliminary experiment (data not shown). Beads with 2 µl of *n*-hexane applied were used as controls. After the beads were left for approximately 12 h to sufficiently volatize the solvent, we started the experiment. We first put a male or a female *E. postfasciatus* approximately 1 cm away from the centre of the bead on the glass dish. After 15 min passed to allow habituation to room temperature ($26 \pm 2^\circ$C), we recorded the behaviour of the individuals for 150 min in total. Until 17 November 2013, the behavioural observation started at 14.30 and ended at 17.00 under dark conditions; we set weevils under flood-lights from 14.30 to 15.00, after which the light was removed. From 18 September 2013 onwards, the behavioural observation started at 12.30 and ended at 15.00; we set weevils under flood-lights from 12.30 to 13.00, after which the light was removed. The light and dark conditions simulated dusk, the time at which *E. postfasciatus* is most active in terms of exploration for both foods and mating. For observing the behaviour of weevils under dark conditions, we used the same electric torch covered with the red plastic film that was adopted in the behavioural experiment in the previous section. The observation time was determined based on evidence that copulation in *E. postfasciatus* is usually initiated within 120 min [22]. Mating behaviour was recorded by visual observation and simultaneously using the video taken with the same digital infrared camera (HDR-CX700 V, SONY, Tokyo, Japan) so as to make certain of indiscernible sexual behaviours of

weevils by checking the scene after the experiment. During the observation, we recorded mounting frequency on the glass bead, accumulated mounting duration on the bead (up to 2000 s per trial) and circumstantial behaviours on the bead based on the definitions described above. We continued the observation when individuals continued mounting after 150 min had passed from the beginning of observation. We then calculated the average duration per mounting on the bead for each trial for subsequent comparison. The trials for each treatment are summarized in table 1.

## 2.5. Statistical analysis

In the behavioural experiment using live insects, we tested the null hypothesis that each stage of the mating event occurred randomly using Fisher's exact test (FET). Provided that all individuals walk randomly in the arena and a male consequently encounters the other male or a female, the expected ratio of homosexual (i.e. male meets male) and heterosexual (i.e. female meets male) encounters is $1:2$. Thus, this ratio was employed as a null hypothesis for detecting deviation in the observed frequency. The observed frequency of post-mounting and pre-copulatory behaviour was then compared with the observed frequency of mounting. We further tested whether the frequency of each behavioural repertoire in the pre-copulatory stage differed from the sex of the male's partner using FET. The duration of courtship and mating trials, defined as the probability of pairing, was compared between sexes using survival analysis. Non-parametric survival curves were plotted based on the Kaplan–Meier method and the difference in the total duration of mating trials was compared between mating pairs using the log-rank test.

For the behavioural assay using the hexane extract, the observed frequencies of each categorized behaviour were compared to those for the null hypothesis using FET. We tested the null hypothesis that individuals walked randomly in the arena and unintentionally encountered the bead. In this case, the null frequency of mounting behaviour would be proportional to that of the experimental trials in each treatment. We then tested the null hypothesis that the post-mounting and pre-copulatory behaviour occurred in proportion to the frequency of mounting. Furthermore, we assessed whether mounting duration varied among treatments by means of a proportional hazards model [39], focusing on leaving tendency (the probability per unit of time that a male or a female leaves the beads, given that a male or a female is still on it). This leaving tendency is assumed to be the product of a baseline leaving tendency and a positive exponential term (i.e. hazard ratio) that represents the joint effect of predefined explanatory factors (=covariates) [40]. The model can be represented as follows:

$$h(t|x) = h_0(t)\exp\left\{\sum_{i=1}^{p}\beta_i x_i\right\},$$

where $h(t|x)$ is the hazard rate, $h_0(t)$ is the baseline bead-leaving tendency (i.e. baseline hazard), $t$ is the time passed because a male or a female started to mount on the beads and $\beta_i$ is the regression coefficient that gives the relative contributions of $p$ covariates $x_i$. The effect of the covariates is given by the value of the hazard ratio. The bead-leaving tendency is reduced if the exponential term is lower than 1 and increased if it is greater than 1. The baseline hazard corresponds to the bead-leaving when all covariates are equal to zero. Here, we took the type of extracts and the gender of individual provided in each trial as covariates. Based on these, we can assess the associated incremental or decremental effect on the leaving tendency of individuals from beads. All the regression coefficients and their variance–covariance matrix were estimated from the data using partial likelihood maximization [41]. All tests were conducted using the R statistical package [42].

# 3. Results

## 3.1. Behavioural experiment using live weevils

In the observation using live weevils, the observed frequency of males mounting males and females did not differ from the expectations of randomness (male:female = 32:41, $p = 0.233$; FET). Differences in the frequency of individuals displaying post-mounting (Stage II) and pre-copulatory behaviour (Stage III) did not significantly deviate from the ratio of males to females mounted by a male (Stage II, male : female = 2 : 10, $p = 0.11$; Stage III, male : female = 26 : 37, $p = 0.86$; FET). Eleven out of 12 males (91.67%) that displayed post-mounting behaviour moved to pre-copulatory processes. In 82.54% (52 out of 63) of pairs, the male displayed pre-copulatory behaviour without obvious post-mounting

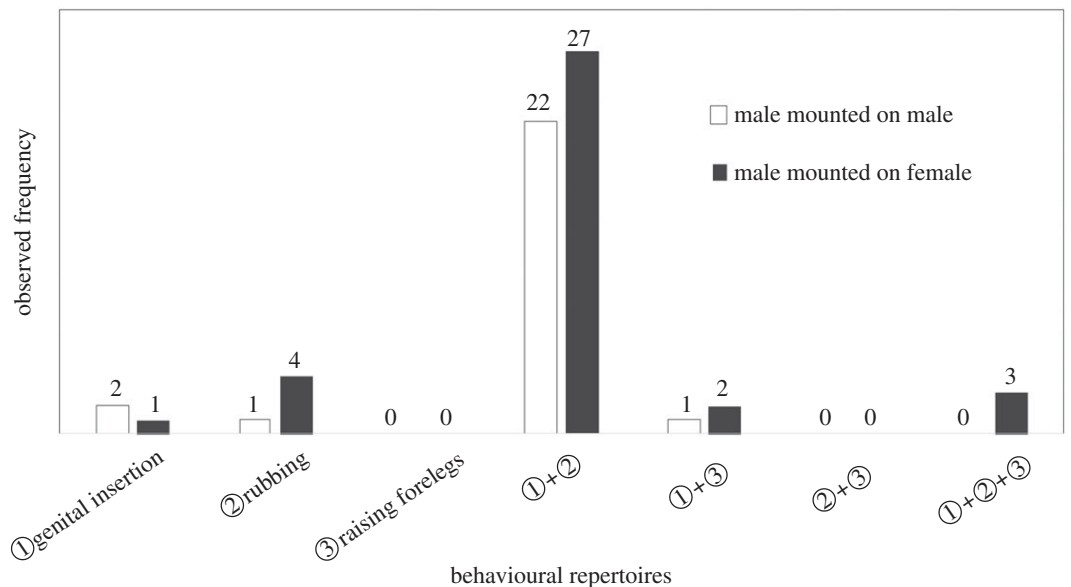

**Figure 2.** Possible combinations of repertoires of pre-copulatory behaviour. Frequencies of defined gestures are overlaid.

**Table 1.** Summary of behavioural assays using the hexane extract. The definition of each stage is described in the Material and methods and figure 1. Note that no male exposed genitalia and exhibited genital insertion on Stage III.

| sex | treatment | $n$ | Stage I | Stage II | Stage III |
|-----|-----------|-----|---------|----------|-----------|
| male | hexane (control) | 40 | 18 | 9 | 0 |
| | male extract | 41 | 26 | 18 | 10 |
| | female extract | 45 | 33 | 23 | 17 |
| female | hexane (control) | 41 | 21 | 6 | — |
| | male extract | 39 | 23 | 14 | — |
| | female extract | 36 | 24 | 12 | — |

behaviour. Of seven possible behavioural repertoires, the most frequent gesture in pre-copulatory behaviour was the strict combination of rubbing and insertion trial in both sexes mounted (figure 2), and there was no significant difference in the frequency of gestures between the sexes ($p = 0.44$; FET). Twenty-five out of 32 (78.13%) males engaged in insertion trials in male–male pairs; however, all the mounting did not persist until the end of the period of observation. For heterosexual (male–female) pairs, by contrast, 18 out of 33 (54.55%) males engaging in pre-copulatory behaviour accomplished successful coupling; others (45.45%) were rejected by the female or stopped mounting. All successfully coupled pairs undertook both rubbing and insertion trials, and several iterations of both gestures were observed until successful coupling. The probability of pairing was significantly higher in male–female pairs than in male–male pairs (figure 3, male–female: $823.27 \pm 128.31$ (mean $\pm$ s.e.m.) s, male–male: $50.04 \pm 7.21$ s.e.m., $\chi^2 = 52.2$, $p \ll 0.0001$).

## 3.2. Behavioural assay using the hexane extracts

The behaviour observed for the weevils on beads is summarized in table 1. The frequency of initial contact behaviour (Stage I) was proportional to the frequency of the beads that have been initially settled in each treatment (i.e. hexane control versus male extract versus female extract) for both sexes ($p = 0.421$ in males, $p = 0.796$ in females; FET). Moreover, no significant deviation in frequency was found between mounting (Stage I) and post-mounting behaviour (Stage II) in either sex ($p = 0.774$ in males, $p = 0.796$ in females; FET). We have to note that no males showed behaviour such as rubbing and/or raising the forelegs, both of which are categorized as Stage III, on beads coated with only hexane solvent (i.e. control), while 10 of 26 and 17 of 33 males displayed either or both of these

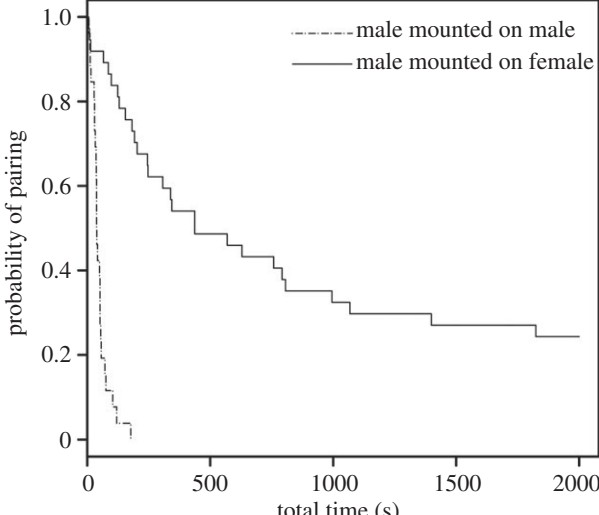

**Figure 3.** Probability of pairing during the stage of pre-copulatory behaviours.

behaviours on beads covered with male and female extract (i.e. treatment); as a consequence, a significant deviation in observed frequency was found between mounting (Stage I) and pre-copulatory (Stage III) behaviour ($p = 0.008$; FET). No male displayed genital insertion against the beads in Stage III (table 1).

To compare bead-leaving tendency based on the Cox proportional hazards model, we first took the gender of weevil and the difference in treatment (i.e. only solvent versus male or female extract) as covariates. A significant decrement in leaving tendency was detected when the gender was male and when the extracted lipid was applied (table 2 and figure 4). Moreover, when the type of extraction (i.e. male versus female extraction) was considered, the interaction between gender (=male) and treatment (=female extract) was significant (table 2). The hazard ratio was less than 1, meaning that the bead-leaving tendency was significantly reduced in this combination.

## 4. Discussion

Although the examination of the composition of chemicals in the extract from the cuticular surface was not the prime focus in the present study, our behavioural experiment using weevil extracts clearly showed that male weevils paid attention to glass beads coated with dissolved compounds and their pre-copulatory behaviour was consequently triggered, suggesting that the males of *E. postfasciatus* perceived some stimulus signals retained on the body surface. However, when examining the frequency of each behavioural stage, the present results could not eliminate the possibility that the weevils unintentionally encountered beads and held them accordingly. Meanwhile, the comparison of averaged dwelling time on the beads revealed that males tended to stay longer on beads than females and that weevils remained longer on beads coated with extracts of body lipids than on beads without extracts (figure 4). As shown in other studies, the extracts are likely to include some semiochemicals, such as long-chain hydrocarbons that are difficult to volatilize, and thus can be perceived using gustatory receivers [43]. Based on the present results, it seems very likely that both male and female weevils perceived chemical stimuli that were embedded in the lipids of congeners, eliciting subsequent behavioural actions. This conjecture is corroborated by evidence that males displayed pre-copulatory behaviour only on the beads coated with extracts, leading to prolonged mounting. These findings are, as far as we are aware, the first report of the effect of lipid components on the body surface on decision-making in *E. postfasciatus*.

The experiment examining mating trials using live weevils also revealed that *E. postfasciatus* males tended to exhibit mounting behaviour irrespective of sex when they encountered a congenerous individual (figure 2). After the mounting, most of them undertook post-mounting and pre-copulatory behaviour and, surprisingly, a genitalia insertion attempt was observed in almost 78% of the males that mounted another male; these behaviours, except the insertion trial of genitalia, were also observed in response to the coated beads mentioned above. Interestingly, female weevils also took Stage II actions on the beads, and furthermore, we confirmed that the females also mounted live

**Table 2.** Cox proportional hazards model results for the model of the probability of leaving bead. Significant *p*-values in italics. (*a*) Results when gender and treatment (presence or absence of extracts) were considered as covariates. The interaction term was not significant and was therefore excluded. (*b*) Results when the types of extracts (i.e. male or female extracts) were considered as covariates.

| covariate | hazard ratio ($\exp^{\beta}$) | z | *p*-values | 95% confidence interval |
|---|---|---|---|---|
| (*a*) | | | | |
| gender | | | | |
| female | 1.00 | – | – | – |
| male | 0.61 | −2.81 | *0.005* | 0.44−0.86 |
| treatment | | | | |
| hexane (control) | 1.00 | – | – | – |
| extract | 0.65 | −2.16 | *0.031* | 0.45−0.96 |
| (*b*) | | | | |
| gender | | | | |
| female | 1.00 | – | – | – |
| male | 1.02 | 0.05 | 0.959 | 0.53−1.93 |
| treatment | | | | |
| hexane (control) | 1.00 | – | – | – |
| male extract | 0.77 | −0.84 | 0.399 | 0.42−1.41 |
| female extract | 1.05 | 0.16 | 0.874 | 0.58−1.91 |
| interaction | | | | |
| male × female extract | 0.43 | −1.99 | *0.047* | 0.18−0.99 |
| male × male extract | 0.57 | −1.26 | 0.208 | 0.24−1.37 |

males and females (because of a lack of sufficient data, we did not include them in the present analysis). These lines of evidence suggest the possibility that males as well as females do not definitively recognize the partner's sex in the period from mounting to the beginning of copulatory processes, as shown in figure 2. The observed SSB in the present experiment can also be observed ubiquitously in various insects and animals [29,30]. SSB is one of the enigmatic issues in evolutionary biology because it is still unclear whether SSB has a certain merit for strengthening reproduction and fitness. In studies on flour beetles, *Tribolium castaneum* (Herbst), male–male coupling may facilitate discarding low-quality sperms, which could improve sperm performance, although further testing is needed [44]. The advantage of sperm dumping, if any, may entail male–male coupling found in the present study; however, this conjecture cannot explain why female–male and female–female coupling behaviours also exist. From a negative point of view, SSB can be regarded as a by-product due to mistaken identity [45]; frequent misdirected coupling might inflate costs of reproduction even in congeners as revealed in a series of studies on reproductive interference [46,47]. We sometimes observed several individuals of *E. postfasciatus* congregated in rearing cages and embracing other individuals. Exploring evolutionary reasons for the maintenance of a series of both SSB and gregarious behaviour should be attempted very carefully while taking various experimental conditions (e.g. field versus laboratory studies, unintended selection in the strain of reared insects) into consideration.

The absence of genital insertion against the beads implies that males realized the partner's identity in the later stage of the copulatory process and consequently quit insemination trials. Because the duration of pre-copulatory behaviour was significantly longer in male–female pairs than in male–male pairs, it appears that males perceive the partner's gender in the pre-copulatory behaviour stage. Thus, it is most probable that the weevils can discern congeners in accordance with components of the waxy extract, but cannot identify the partner's gender solely based on them. Interestingly, our data provided evidence that successful insertion (and possible insemination) was finally rejected by females in approximately. 45% of males. Males of *E. postfasciatus* are known to possess spines on the surface of the endophallus, and the shape of the spines significantly affects female remating propensity [48]. As revealed in a polyandrous bean beetle, *Callosobruchus maculatus*, the males of which also have

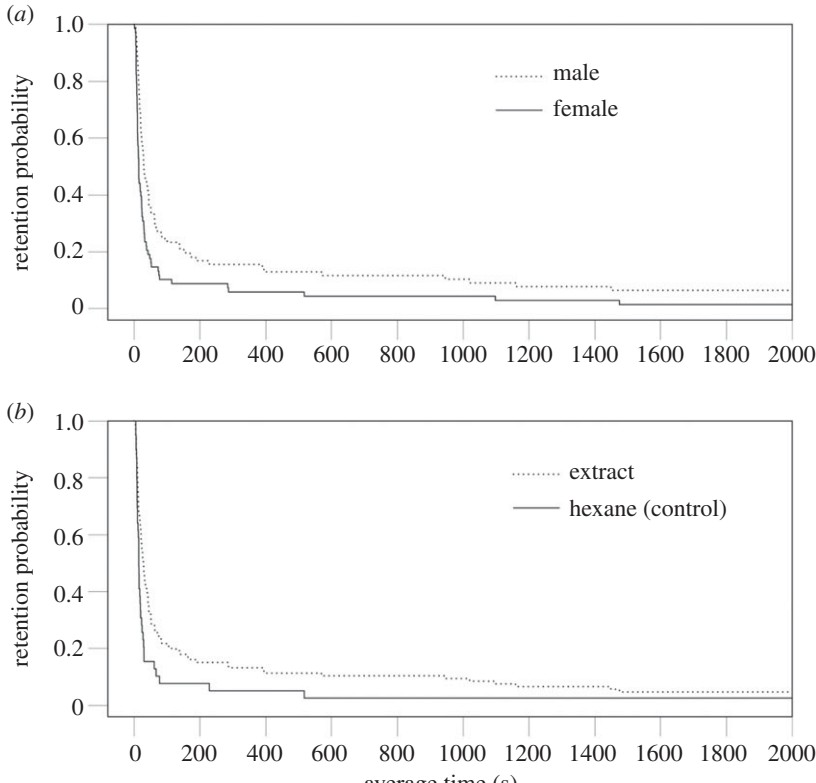

**Figure 4.** Probability of retention on beads (see the text for details) influenced by (*a*) gender of weevils and (*b*) difference in treatment.

genital spines that are harmful to females [49,50], the females of *E. postfasciatus* suffer damage from an everted endophallus in their bursa and consequently refuse remating for approximately one week after copulation [21,23]. Therefore, for females, it is essential to choose the partner before suffering damage from unwilling copulation. The present results suggest the possibility that the females do not choose the partner before the steps of pre-copulatory process, but discriminate during the initial insertion of male genitalia. Most males, however, stopped mounting in the precursory period of the pre-copulatory processes when the partner was male (figure 3); therefore, it is probable that they make the decision to extend copulation by making use of the partner's reactions and/or specific traits, if any, of the partner.

In general, while there is a line of empirical studies suggesting that chemical cues such as cuticular hydrocarbons play an important role in promoting preferred mating and finding foods in various species of insects (e.g. [51]), other perceptual signals, such as visual, tactile and auditory stimuli, are also indispensable in identifying the antecedents of other individuals [52,53]. Interestingly, both males and females of *E. postfasciatus* are equipped with stridulatory files on the elytron and plectrums on the abdomen, and they are known to emit sound using these apparatuses when they are engaging in mating or suffer physical (stressful) stimuli [54,55]; however, the practical roles of the sound in the sequence of mating behaviour have not been explained fully. It has also been shown that echemes in the sounds emitted by females are substantially longer than those emitted by males [56]. As shown in other insects, it is most probable that *E. postfasciatus* leverages different stimuli simultaneously or sequentially for the recognition of the other sex and facilitating insemination (e.g. [57]).

Our study demonstrated that weevils responded to extracted reagents from body lipids and modified their decisions accordingly. This finding implies that extracts from the body surface could make weevils stay longer on a substrate that is coated with them. Regarding monitoring weevils in the wild, despite a lack of attractiveness from afar, the extracted reagents might be applied to make the weevils stay longer around preferable spots. In fact, although the weevils tend to be attracted to specific wavelengths of light [19,58], no effective method has been established to keep weevils inside the trap. It is well known that both specific colours and pheromones exert attractiveness on various insects [59], and these cues have been exploited for the development of traps that are effective against monitoring pest species [60]. In

consideration of our findings, it is worthwhile to conduct an additional treatment such as spreading weevil extracts inside the apparatus in order to ascertain whether the number of weevils captured increases significantly. The assessment of extracts using gas chromatography and mass spectrometry will serve to identify active ingredients for attractiveness.

Ethics. The present experiments were performed in accordance with the guidelines for ethological studies from the Japan Ethological Society.

Data accessibility. The datasets used in the present study are available at Dryad: http://dx.doi.org/10.5061/dryad.45ng8 [61].

Authors' contributions. M.I., N.K. and H.T. designed the experiments. M.I. carried out the experiments. H.T. and M.I. completed the data analysis and drafted the manuscript, and N.K. assisted with experimental work and helped draft the manuscript. All authors gave final approval for publication.

Competing interests. The authors declare no competing interests.

Funding. Japan Society for the Promotion of Science research grant was awarded to H.T. (nos. 25304014; 17H03722; 17K20068) and N.K. (no. 26450067). University of the Ryukyus Research Incentive Grant for KAKENHI Acquisition was awarded to H.T. (no. 184-1).

Acknowledgements. We cordially thank all members of the Laboratory of Entomology at University of the Ryukyus for their thoughtful discussions and encouragement. We also thank D. Haraguchi and T. Matsuyama (OPPPC) for kindly supplying weevils for the present experiment, and T. Sasaki (University of the Ryukyus) for providing a dark room.

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
