## [Reviewer comments · Royal Society Open Science]

Review History

RSOS-172469.R0 (Original submission)

Review form: Reviewer 1

Is the manuscript scientifically sound in its present form?

Yes

Are the interpretations and conclusions justified by the results?

No

Is the language acceptable?

Yes

Is it clear how to access all supporting data?

No

Do you have any ethical concerns with this paper?

No

Have you any concerns about statistical analyses in this paper?

No

Recommendation?

Major revision is needed (please make suggestions in comments)

Comments to the Author(s)

This manuscript presents research on the mating behavior of a weevil species and attempts to characterize the importance of cuticular compounds (specifically hydrocarbons) on mating behavior. While narrow in scope, the paper is well written and the statistical analyses are sound (but see exception below).

I have serious concerns about the author's definition of courtship behavior. The authors make the statement that 'courtship behavior' is inappropriate to describe male-male mounting behaviors. However, a behavior does not depend on the outcome (copulation or insertion of genitalia), it is the sequence of events that are predictable and repeatable (per the ethogram in Fig 1, stages 1-3). This phrase needs to be removed and the authors should utilize a comprehensive term that includes the same-sex mountings. Considering the high percentage of male-male pairs (78%) observed, the authors need to revise places in the analysis where male-male mountings were excluded for instances of determining behavior. As the authors mention, one of their objectives is to determine mechanisms of gender identification, specifically as it pertains to the mating sequence. From the places where it was included, the data suggests that males do not distinguish between genders (pg 5) but rather the distinction is upon females.

Specific comments:

Pg 3, line 15: Rephrase to state "volatilization of beads occurs for X hours"

Pg 3, line 26, replace judgement with observation.

Pg 3 - how was video used? please describe

Pg 4, ln10: same-sex riding behavior? use same-sex mounting behavior

Suggest putting observation on "Behavioral experiment using alive weevils" before the section using hexane extracts

pg 4, line 52: did you compare within sexes for duration of courtship and mating trials? Include.

Pg 5, line 1-4: clarify that the treatments evaluated here are the hydrocarbon extract vs. the hexane control

Discussion

The authors should expand the discussion to compare their results to the broader literature, specifically other instances where males do not differentiate gender. What are the implications of this research?

Review form: Reviewer 2**Is the manuscript scientifically sound in its present form?**

Yes

Are the interpretations and conclusions justified by the results?

Yes

Is the language acceptable?

Yes

Is it clear how to access all supporting data?

Not Applicable

Do you have any ethical concerns with this paper?

No

Have you any concerns about statistical analyses in this paper?

No

Recommendation?

Accept with minor revision (please list in comments)

Comments to the Author(s)

This is a behavioral study that investigated a sequence of mating of the West Indian sweet potato weevil, *Euscepes postfasciatus*, a notorious invasive pest in sweet potato fields in southwestern islands of Japan. The authors described under-laboratory observations of mating behaviors of the weevil by using both live insects and crude solvent extracts containing waxy components on their body surface in glass dishes. The authors also analyzed their data by using the Cox proportional hazard model and clearly showed that male individuals were significantly arrested by female-body-derived hexane extracts. Overall, I found their experiments and analyses are well-done. However, I think the manuscript has a couple of issues to be addressed before publication. My major concern is that the authors described almost nothing about a potential value of their findings, that is the function of body surface waxy compounds in mating of the weevil, for pest managements, nevertheless they stated in Abstracts and Introduction that "The present study would considerably aid the identification of effective attractants for the weevils and thus for the establishment of successful pest management (P2, L25)". I agree that the waxy compounds found in this study are not volatile. Moreover, this insect loses flight ability (P6, L44). Thus I imagine it is difficult to use these chemical stimuli for monitoring trap lures or improvements of them. I would like the authors to show some idea to apply their insight to pest management programs of this serious pest weevil.

My other comments are as follows;

1. P1, L28

mating -> mate

2. P2, L44

Kohama and Sugiyama (2000) -> please revise this to a proper citation style.

3. P3, L4

the male grasps the other individual... -> according to the data in Table 1, female weevils also show the behavior of Stage II. Is there no difference between female and male behaviors? Please clarify.

4. P3, L22

There is no descriptions about the age (12-17-d-old as individuals used in bioassays?) and mating status of weevils used for extraction, which is critical in the present experiment. Also, timing of

extraction is generally essential for chemical stimuli used in sexual communications. Please clearly describe them.

5. P3, L26

These extracts were transferred to a clean glass vial and evaporated to 30 μ l -> These extracts were transferred to a clean glass vial and evaporated to 30 μ l under nitrogen stream (? Or under vacuum? using an evaporator?)

6. P3, L34

ca. 4 individuals -> ca. 4-individual-equivalent)

7. P3, L35

covered -> applied

8. P3, L43

after which the light was removed -> how did you observe insects under a dark photoperiod? Using a red light as the next observation?

9. P4, L9

same-sex riding -> intrasexual mounting

10. P4, L45

reproductive -> mating

11. P5, L7

However, in males, no individual showed behaviour such as rubbing and/or raising the forelegs in the control experiment (only hexane solvent), while 10 of 26 and 17 of 33 males displayed either or both of these behaviour on beads covered with male and female extract, respectively; as a result, the difference in frequency was significant between mounting and pre-copulatory behaviour ($P = 0.008$; FET) -> this sentence is hard to be understand; how did you calculate $P=0.008$?

12. P5, L23

(male:female=32:41, $P=1$; FET) -> is male:female=32:41 an expected number, which is identical to an observed number? If not, $P=1$ is error.

13. P5, L45

attraction -> behavioral

14. P6, L3

The experiment examining mating trials using live weevils also revealed that *E. postfasciatus* males tended to exhibit mounting behaviour irrespective of sex when they encountered a congenerous individual. -> I agree that this observation is unusual and interesting. Is this often found in weevils or other beetles? Alternatively, is there any possibilities that such the behavior is triggered by laboratory manipulations? For example, the behavior was investigated in a very limited space (4.7 mm in diameter), and the insects are reared under laboratory for a long term (nearly 10 years), which may have driven unintentional selections such as a kin selection. I would like the authors to discuss more about this topic.

15. Figure 3

There is no label in the Y-axis. Please show it.

Decision letter (RSOS-172469.R0)

30-Jan-2018

Dear Dr Tatsuta:

Manuscript ID RSOS-172469 entitled "When a male perceives a female: the effect of waxy components on the body surface on decision-making in the invasive pest weevil" which you submitted to Royal Society Open Science, has been reviewed. The comments from reviewers are included at the bottom of this letter.

In view of the criticisms of the reviewers, the manuscript has been rejected in its current form. However, a new manuscript may be submitted which takes into consideration these comments.

Please note that resubmitting your manuscript does not guarantee eventual acceptance, and that your resubmission will be subject to peer review before a decision is made.

Your resubmitted manuscript should be submitted by 30-Jul-2018. If you are unable to submit by this date please contact the Editorial Office.

Please note that Royal Society Open Science will introduce article processing charges for all new submissions received from 1 January 2018. Charges will also apply to papers transferred to Royal Society Open Science from other Royal Society Publishing journals, as well as papers submitted as part of our collaboration with the Royal Society of Chemistry (<http://rsos.royalsocietypublishing.org/chemistry>). If your manuscript is submitted and accepted for publication after 1 Jan 2018, you will be asked to pay the article processing charge, unless you request a waiver and this is approved by Royal Society Publishing. You can find out more about the charges at <http://rsos.royalsocietypublishing.org/page/charges>. Should you have any queries, please contact openscience@royalsociety.org.

on behalf of Dr Alexander Ophir (Associate Editor) and Kevin Padian (Subject Editor)
openscience@royalsociety.org

Associate Editor Comments to Author (Dr Alexander Ophir):

Dear Dr. Tatsuta,

I have now received the comments from two expert reviewers of your manuscript. As you will see there were some very positive comments regarding your paper and I believe there is some important value to your study. However, both reviewers raised some important concerns, including that you have not discussed the importance of this work in a way that will resonate with a broad audience. Reviewer 1 also raises a very important conceptual problem with your paper: that your characterization of courtship behavior is unjustified. Indeed, I agree with their argument that a behavior does not depend on the outcome (copulation or insertion of genitalia), it is a sequence of events that is predictable and repeatable. The omission of some of your behavioral data (male-male mounting) also posed an issue if your goal was to identify chemical cues for intraspecific gender identification. Your interpretations will require adjustment. Finally, as I mentioned, both reviewers indicated that your manuscript is narrow in scope, a concern I had originally. Their confirmation of this point has convinced me that your paper would be more appropriate in a Taxon-specific journal such as *Insect Behavior*.

As a result, unfortunately, I must recommend rejection of your paper for publication in *RSOS*.

Subject Editor Comments to Author:

I appreciated the comments of one reviewer and the AE that this study is narrow in its scope; however, that is not a criterion for *RSOS*, and so this issue must be discounted. However there are some major issues with the interpretation raised by the reviewers and so we will allow resubmission if these are addressed. thanks for your submission.

Reviewers' Comments to Author:

Reviewer: 1

Comments to the Author(s)

This manuscript presents research on the mating behavior of a weevil species and attempts to characterize the importance of cuticular compounds (specifically hydrocarbons) on mating behavior. While narrow in scope, the paper is well written and the statistical analyses are sound (but see exception below).

I have serious concerns about the author's definition of courtship behavior. The authors make the statement that 'courtship behavior' is inappropriate to describe male-male mounting behaviors. However, a behavior does not depend on the outcome (copulation or insertion of genitalia), it is the sequence of events that are predictable and repeatable (per the ethogram in Fig 1, stages 1-3). This phrase needs to be removed and the authors should utilize a comprehensive term that includes the same-sex mountings. Considering the high percentage of male-male pairs (78%) observed, the authors need to revise places in the analysis where male-male mountings were excluded for instances of determining behavior. As the authors mention, one of their objectives is to determine mechanisms of gender identification, specifically as it pertains to the mating sequence. From the places where it was included, the data suggests that males do not distinguish between genders (pg 5) but rather the distinction is upon females.

Specific comments:

Pg 3, line 15: Rephrase to state "volatilization of beads occurs for X hours"

Pg 3, line 26, replace judgement with observation.

Pg 3 - how was video used? please describe

Pg 4, ln10: same-sex riding behavior? use same-sex mounting behavior

Suggest putting observation on "Behavioral experiment using alive weevils" before the section using hexane extracts

pg 4, line 52: did you compare within sexes for duration of courtship and mating trials? Include.

Pg 5, line 1-4: clarify that the treatments evaluated here are the hydrocarbon extract vs. the hexane control

Discussion

The authors should expand the discussion to compare their results to the broader literature, specifically other instances where males do not differentiate gender. What are the implications of this research?

Reviewer: 2

Comments to the Author(s)

This is a behavioral study that investigated a sequence of mating of the West Indian sweet potato weevil, *Euscepes postfasciatus*, a notorious invasive pest in sweet potato fields in southwestern islands of Japan. The authors described under-laboratory observations of mating behaviors of the weevil by using both live insects and crude solvent extracts containing waxy components on their body surface in glass dishes. The authors also analyzed their data by using the Cox proportional hazard model and clearly showed that male individuals were significantly arrested by female-body-derived hexane extracts. Overall, I found their experiments and analyses are well-done. However, I think the manuscript has a couple of issues to be addressed before publication. My major concern is that the authors described almost nothing about a potential value of their findings, that is the function of body surface waxy compounds in mating of the weevil, for pest managements, nevertheless they stated in Abstracts and Introduction that "The present study would considerably aid the identification of effective attractants for the weevils and thus for the establishment of successful pest management (P2, L25)". I agree that the waxy compounds found in this study are not volatile. Moreover, this insect loses flight ability (P6, L44). Thus I imagine it is difficult to use these chemical stimuli for monitoring trap lures or improvements of them. I would like the authors to show some idea to apply their insight to pest management programs of this serious pest weevil.

My other comments are as follows;

1. P1, L28

mating -> mate

2. P2, L44

Kohama and Sugiyama (2000) -> please revise this to a proper citation style.

3. P3, L4

the male grasps the other individual... -> according to the data in Table 1, female weevils also show the behavior of Stage II. Is there no difference between female and male behaviors? Please clarify.

4. P3, L22

There is no descriptions about the age (12-17-d-old as individuals used in bioassays?) and mating status of weevils used for extraction, which is critical in the present experiment. Also, timing of

extraction is generally essential for chemical stimuli used in sexual communications. Please clearly describe them.

5. P3, L26

These extracts were transferred to a clean glass vial and evaporated to 30 μ l -> These extracts were transferred to a clean glass vial and evaporated to 30 μ l under nitrogen stream (? Or under vacuum? using an evaporator?)

6. P3, L34

ca. 4 individuals -> ca. 4-individual-equivalent)

7. P3, L35

covered -> applied

8. P3, L43

after which the light was removed -> how did you observe insects under a dark photoperiod? Using a red light as the next observation?

9. P4, L9

same-sex riding -> intrasexual mounting

10. P4, L45

reproductive -> mating

11. P5, L7

However, in males, no individual showed behaviour such as rubbing and/or raising the forelegs in the control experiment (only hexane solvent), while 10 of 26 and 17 of 33 males displayed either or both of these behaviour on beads covered with male and female extract, respectively; as a result, the difference in frequency was significant between mounting and pre-copulatory behaviour ($P = 0.008$; FET) -> this sentence is hard to be understand; how did you calculate $P=0.008$?

12. P5, L23

(male:female=32:41, $P=1$; FET) -> is male:female=32:41 an expected number, which is identical to an observed number? If not, $P=1$ is error.

13. P5, L45

attraction -> behavioral

14. P6, L3

The experiment examining mating trials using live weevils also revealed that *E. postfasciatus* males tended to exhibit mounting behaviour irrespective of sex when they encountered a congenerous individual. -> I agree that this observation is unusual and interesting. Is this often found in weevils or other beetles? Alternatively, is there any possibilities that such the behavior is triggered by laboratory manipulations? For example, the behavior was investigated in a very limited space (4.7 mm in diameter), and the insects are reared under laboratory for a long term (nearly 10 years), which may have driven unintentional selections such as a kin selection. I would like the authors to discuss more about this topic.

15. Figure 3

There is no label in the Y-axis. Please show it.

Author's Response to Decision Letter for (RSOS-172469.R0)

See Appendix A.

RSOS-181542.R0

Review form: Reviewer 2

Is the manuscript scientifically sound in its present form?

Yes

Are the interpretations and conclusions justified by the results?

Yes

Is the language acceptable?

Yes

Is it clear how to access all supporting data?

Yes

Do you have any ethical concerns with this paper?

No

Have you any concerns about statistical analyses in this paper?

No

Recommendation?

Accept with minor revision (please list in comments)

Comments to the Author(s)

I found that the revised manuscript is substantially improved following the comments from the reviewers. I would like to request a couple of additional revisions.

P9, L36 and some other places
extracted reagents -> extracts

P10, L26; P11, L35
alive -> live

P12, L10
30 μ l -> ca. 30 μ l

P12, L11
2 individuals/ μ l -> ca. 2 individual equivalents/ μ l

P12, L15
4-individual-equivalent -> 4 individual equivalents

P12, L45

the expected ratio of male-to-female encounters is 2:1 -> the expected ratio of homosexual and heterosexual encounters is 1:2. (? Please confirm this sentence again.)

P15, L24

This finding implies that extracted lipid composition could make weevils stay longer on a substrate that is coated with extracted compounds from the body surface of the weevils -> This finding implies that extracts from the body surface could make weevils stay longer on a substrate that is coated with them

Review form: Reviewer 3 (Clement Akotsen-Mensah)

Is the manuscript scientifically sound in its present form?

No

Are the interpretations and conclusions justified by the results?

No

Is the language acceptable?

Yes

Is it clear how to access all supporting data?

Not Applicable

Do you have any ethical concerns with this paper?

No

Have you any concerns about statistical analyses in this paper?

I do not feel qualified to assess the statistics

Recommendation?

Major revision is needed (please make suggestions in comments)

Comments to the Author(s)

The text can be reduced significantly since most of the statements were overly described and can be shortened.

Decision letter (RSOS-181542.R0)

17-Dec-2018

Dear Dr Tatsuta

On behalf of the Editor, I am pleased to inform you that your Manuscript RSOS-181542 entitled "When a male perceives a female: the effect of waxy components on the body surface on decision-

making in the invasive pest weevil" has been accepted for publication in Royal Society Open Science subject to minor revision in accordance with the referee suggestions. Please find the referees' comments at the end of this email.

The reviewers and Subject Editor have recommended publication, but also suggest some minor revisions to your manuscript. Therefore, I invite you to respond to the comments and revise your manuscript.

- Ethics statement

- Data accessibility

If you wish to submit your supporting data or code to Dryad (<http://datadryad.org/>), or modify your current submission to dryad, please use the following link:
<http://datadryad.org/submit?journalID=RSOS&manu=RSOS-181542>

- Competing interests

- Authors' contributions

- Acknowledgements

- Funding statement

Because the schedule for publication is very tight, it is a condition of publication that you submit the revised version of your manuscript before 26-Dec-2018. Please note that the revision deadline will expire at 00.00am on this date. If you do not think you will be able to meet this date please let me know immediately.

on behalf of Dr Alexander Ophir (Associate Editor) and Kevin Padian (Subject Editor)
openscience@royalsociety.org

Associate Editor Comments to Author (Dr Alexander Ophir):

Dear Dr. Tatsuta,

I have once again received the reviews from two reviewers. Reviewer 1 was largely satisfied with your revised submission with a few very easily addressable points that you should incorporate. The second reviewer did not provide many specifics, but felt that you should shorten your paper. This could be achieved by being more succinct in your discussion points and by potentially reducing some unnecessary references. The second reviewer also felt that your study would have been more compelling if you had incorporated gas-chromatography and mass spectrometry to obtain pure compounds and used that instead of just the extracts. Perhaps you can briefly address this point in your discussion section? These points aside, you have done a nice job of addressing the concerns of the previous reviews and I appreciate your efforts on your manuscript. I look forward to receiving a finalized version of this paper.

Best
Alex Ophir

Reviewer comments to Author:
Reviewer: 2

Comments to the Author(s)

I found that the revised manuscript is substantially improved following the comments from the reviewers. I would like to request a couple of additional revisions.

P9, L36 and some other places
extracted reagents -> extracts

P10, L26; P11, L35
alive -> live

P12, L10
30 μ l -> ca. 30 μ l

P12, L11
2 individuals/ μ l -> ca. 2 individual equivalents/ μ l

P12, L15

4-individual-equivalent -> 4 individual equivalents

P12, L45

the expected ratio of male-to-female encounters is 2:1 -> the expected ratio of homosexual and heterosexual encounters is 1:2. (? Please confirm this sentence again.)

P15, L24

This finding implies that extracted lipid composition could make weevils stay longer on a substrate that is coated with extracted compounds from the body surface of the weevils -> This finding implies that extracts from the body surface could make weevils stay longer on a substrate that is coated with them

Reviewer: 3

Comments to the Author(s)

The text can be reduced significantly since most of the statements were overly described and can be shortened.

Author's Response to Decision Letter for (RSOS-181542.R0)

See Appendix B.

Decision letter (RSOS-181542.R1)

09-Jan-2019

Dear Dr Tatsuta,

I am pleased to inform you that your manuscript entitled "When a male perceives a female: the effect of waxy components on the body surface on decision-making in the invasive pest weevil" is now accepted for publication in Royal Society Open Science.

Kind regards,
Andrew Dunn
Senior Publishing Editor
Royal Society Open Science Editorial Office
Royal Society Open Science
openscience@royalsociety.org

on behalf of Dr Alexander Ophir (Associate Editor) and Kevin Padian (Subject Editor)
openscience@royalsociety.org

Appendix A

Reply to reviewers

We greatly appreciate many thoughtful comments and suggestions. We have tried to respond to queries and suggestions one by one with religious care. The corrected parts and words as well as replies to reviewers are indicated using letters of brown colour.

>Associate editor:

I have now received the comments from two expert reviewers of your manuscript. As you will see there were some very positive comments regarding your paper and I believe there is some important value to your study. However, both reviewers raised some important concerns, including that you have not discussed the importance of this work in a way that will resonate with a broad audience. Reviewer 1 also raises a very important conceptual problem with your paper: that your characterization of courtship behavior is unjustified. Indeed, I agree with their argument that a behavior does not depend on the outcome (copulation or insertion of genitalia), it is a sequence of events that is predictable and repeatable. The omission of some of your behavioral data (male-male mounting) also posed an issue if your goal was to identify chemical cues for infraspecific gender identification. Your interpretations will require adjustment. Finally, as I mentioned, both reviewers indicated that your manuscript is narrow in scope, a concern I had originally. Their confirmation of this point has convinced me that your paper would be more appropriate in a Taxon-specific journal such as *Insect Behavior*.

-Reply

We appreciate your comments and suggestions. We tried to improve the manuscript in accordance with the advice of two experts and added

discussion about the same-sex sexual behaviour (SSB).

>Subject Editor Comments to Author:

I appreciated the comments of one reviewer and the AE that this study is narrow in its scope; however, that is not a criterion for RSOS, and so this issue must be discounted. However there are some major issues with the interpretation raised by the reviewers and so we will allow resubmission if these are addressed. thanks for your submission.

-Reply

We are really grateful for your thoughtful suggestion. We tried to broaden the scope of topics we dealt with in the present study according to the suggestions by two experts, especially for the intriguing issue of same-sex sexual behaviours (SSB) and the possible usage of extracted reagents in monitoring weevils.

Reviewers' Comments to Author:

Reviewer: 1

I have serious concerns about the author's definition of courtship behavior. The authors make the statement that 'courtship behavior' is inappropriate to describe male-male mounting behaviors. However, a behavior does not depend on the outcome (copulation or insertion of genitalia), it is the sequence of events that are predictable and repeatable (per the ethogram in Fig 1, stages 1-3). This phrase needs to be removed and the authors should utilize a comprehensive term that includes the same-sex mountings. Considering the high percentage of male-male pairs (78%) observed, the authors need to revise places in the analysis where male-male mountings were excluded for instances of determining behavior. As the authors mention, one of their objectives is to determine mechanisms of gender identification, specifically as it pertains to the mating sequence. From the places where it was included, the data

suggests that males do not distinguish between genders (pg 5) but rather the distinction is upon females.

-Reply

Thank you very much for valuable suggestions. We totally agree with the above opinion. We have learned the same-sex sexual behaviours (SSB) from some recent literatures and realised that SSB is common in some insect species, while it is controversial to determine whether SSB is merely a by-product and is non-adaptive. We updated our knowledge on SSB, especially based on the magnificent review of Scharf & Martin (2013: *Behav Ecol Sociobiol* 67). We added explanation for SSB in Introduction (L37-39) and in the section 3.2 in Materials and Methods.

Specific comments:

Pg 3, line 15: Rephrase to state "volatilization of beads occurs for X hours"

-Reply

We rephrased as "After the beads were left for ca.12 hours to sufficiently volatilize the solvent..."

Pg 3, line 26, replace judgement with observation.

-Reply

We replaced the word.

Pg 3 - how was video used? please describe

-Reply

Since we were not sometimes really confident of our judgement of weevil behaviours on beads that were hard to see under the dark condition. To make sure the judgement, we recorded the behaviours using video and made certain of the judgement by looking at it after the experiments. A phrase explaining these was inserted in the texts (section 3.4 in Materials and Methods).

Pg 4, ln10: same-sex riding behavior? use same-sex mounting behavior

-Reply

Ref#2 also suggested a bit different wording. We described as “same-sex sexual behaviour (SSB)” as defined in the section of 3.2 in Materials and Methods.

Suggest putting observation on "Behavioral experiment using alive weevils" before the section using hexane extracts

-Reply

We moved the section on “Behavioural experiment using alive weevils” as suggested. Accordingly, we altered the order of appearance of sections in the Results. In addition, we exchanged the order of statistical explanations in section 3.5 in Materials and Methods so as to fit the order of appearance.

pg 4, line 52: did you compare within sexes for duration of courtship and mating trials? Include.

-Reply

We noticed that the previous description was inappropriate; precisely, we compared between mating pairs (male-male vs. male-female pair), not sexes. We corrected the sentence so as to clarify the meaning.

Pg 5, line 1-4: clarify that the treatments evaluated here are the hydrocarbon extract vs. the hexane control

-Reply

Here we compared whether the frequency of contact behavior was proportional to the frequency of the beads that have been initially settled in each treatment (i.e. hexane control vs. male extract vs. female extract) in each sex. Indeed, the initial sentence seems confusing, so we changed it to clarify the meaning. Please see section 4.2 in Results.

Discussion

The authors should expand the discussion to compare their results to the broader literature, specifically other instances where males do not differentiate gender. What are the implications of this research?

-Reply

We added general discussions concerning SSB, with an example of flour beetle. We also mentioned the conditions of the experiment, which was pointed out by ref#2; however, because we did not know the details of hidden background on the genetic nature of reared weevils, we just raised the plausible items that might affect the sexual behaviours (the second paragraph in Discussion).

Reviewer: 2

However, I think the manuscript has a couple of issues to be addressed before publication. My major concern is that the authors described almost nothing about a potential value of their findings, that is the function of body surface waxy compounds in mating of the weevil, for pest managements, nevertheless they stated in Abstracts and Introduction that “The present study would considerably aid the identification of effective attractants for the weevils and thus for the establishment of successful pest management (P2, L25)”. I agree that the waxy compounds found in this study are not volatile. Moreover, this insect loses flight ability (P6, L44). Thus I imagine it is difficult to use these chemical stimuli for monitoring trap lures or improvements of them. I would like the authors to show some idea to apply their insight to pest management programs of this serious pest weevil.

-Reply

Thank you for pointing out very substantial issue. We addressed our idea to improve effectiveness of trapping weevils in the last paragraph in Discussion.

My other comments are as follows;

1. P1, L28

mating -> mate

-Reply

We corrected it.

2. P2, L44

Kohama and Sugiyama (2000) -> please revise this to a proper citation style.

-Reply

We corrected it so as to fit the style of RSOS.

3. P3, L4

the male grasps the other individual... -> according to the data in Table 1, female weevils also show the behavior of Stage II. Is there no difference between female and male behaviors? Please clarify.

-Reply

Thank you for the important comments. Most of studies investigating mating behaviors using the weevils have mainly focused on the behaviour of males, not of females. We found in the present study that females also took actions categorised in Stage II when beads were provided. There seems no marked difference in behaviours between the sexes. We added the description in the texts as “We note here that we found the female also took actions very similar to those taken by males categorised in Stage II in the present experiment (also see Table 1). ”.

4. P3, L22

There is no descriptions about the age (12-17-d-old as individuals used in bioassays?) and mating status of weevils used for extraction, which is

critical in the present experiment. Also, timing of extraction is generally essential for chemical stimuli used in sexual communications. Please clearly describe them.

-Reply

As described in the section 3.1, all individuals used in behavioural experiments were 12-17-day-old virgin weevils. These weevils can be regarded as well sexually matured; thus are suitable for assays of sexual communications. To clarify the contexts, we added the description in the texts as “We used sexually matured 12-17-day-old virgin weevils were used for the extraction of cuticular lipids.”.

5. P3, L26

These extracts were transferred to a clean glass vial and evaporated to 30 μ l -> These extracts were transferred to a clean glass vial and evaporated to 30 μ l under nitrogen stream (? Or under vacuum? using an evaporator?)

-Reply

We used nitrogen stream for the evaporation, so added the explanation in texts.

6. P3, L34

ca. 4 individuals -> ca. 4-individual-equivalent)

-Reply

We corrected it as suggested.

7. P3, L35

covered -> applied

-Reply

We corrected it.

8. P3, L43

after which the light was removed -> how did you observe insects under a dark photoperiod? Using a red light as the next observation?

-Reply

Yes, we used the same red light appeared in the next section. Because we exchanged behavioural assay and experiment using alive insects, we added the explanation of torch in newly numbered section 3.4 (behavioural assay section).

9. P4, L9

same-sex riding -> intrasexual mounting

-Reply

We termed as “same-sex-sexual behaviour (SSB)” as proposed by Bailey & Zuk (2009) and Scharf & Martin (2013). This indication is related to the main criticism by Ref#1.

10. P4, L45

reproductive -> mating

-Reply

We corrected it.

11. P5, L7

However, in males, no individual showed behaviour such as rubbing and/or raising the forelegs in the control experiment (only hexane solvent), while 10 of 26 and 17 of 33 males displayed either or both of these behaviour on beads covered with male and female extract, respectively; as a result, the difference in frequency was significant between mounting and pre-copulatory behaviour ($P = 0.008$; FET) -> this sentence is hard to be understand; how did you calculate $P=0.008$?

-Reply

We compared frequencies in Stage I and those in Stage III in males. In order to clarify the meaning, we rephrased the corresponding part as follows:

“We have to note that no males showed behaviour such as rubbing and/or raising the forelegs, both of which are categorised as Stage III, on beads coated with only hexane solvent (i.e. control), while 10 of 26 and 17 of 33 males displayed either or both of these behaviours on beads covered with male and female extract (i.e. treatment); as a consequence, a significant deviation in observed frequency was found between mounting (Stage I) and pre-copulatory (Stage III) behaviour ($P = 0.008$; FET).”

12. P5, L23

(male:female=32:41, $P=1$; FET) -> is male:female=32:41 an expected number, which is identical to an observed number? If not, $P=1$ is error.

-Reply

We appreciate the important comments you made about the problem. As pointed out, the P-value was definitely incorrect; this was because we used expected proportion, not expected frequency. We have done FET and put the correct P-value ($P = 0.233$) in the text.

13. P5, L45

attraction -> behavioral

-Reply

We corrected it.

14. P6, L3

The experiment examining mating trials using live weevils also revealed that *E. postfasciatus* males tended to exhibit mounting behaviour irrespective of sex when they encountered a congenerous individual. -> I

agree that this observation is unusual and interesting. Is this often found in weevils or other beetles? Alternatively, is there any possibilities that such the behavior is triggered by laboratory manipulations? For example, the behavior was investigated in a very limited space (4.7 mm in diameter), and the insects are reared under laboratory for a long term (nearly 10 years), which may have driven unintentional selections such as a kin selection. I would like the authors to discuss more about this topic.

-Reply

Thank you for intriguing suggestions. This issue is quite interesting and studies revealed that same-sex-sexual behavior (SSB) is also found in a flour beetle *Tribolium castaneum* (Levan *et al.* 2008: *J Evol Biol* 22). We added the details of SSB in other animals as well as experimental conditions in Discussion section. Because neither SSB found in wild condition nor genetic background in the lineages of reared weevils has been provided, we just raised unintended matters that might affect mating behaviours in the second paragraph in Discussion. The diameter of a glass Petri dish provided in behavioural assay using hexane extracts was 27mm, so 4.7mm is probably a misinterpretation.

15. Figure 3

There is no label in the Y-axis. Please show it.

-Reply

We added labels in both X- and Y-axis.

Further comments from authors:

We corrected inappropriate descriptions that have not been pointed out in the previous peer review. These are designated as red letters.

Appendix B

Reply to reviewers

We again appreciate many thoughtful comments and suggestions. We have tried to respond to queries and suggestions one by one with religious care. The corrected parts and words as well as replies to reviewers are indicated using letters of brown colour.

>Associate editor:

I have once again received the reviews from two reviewers. Reviewer 1 was largely satisfied with your revised submission with a few very easily addressable points that you should incorporate. The second reviewer did not provide many specifics, but felt that you should shorten your paper. This could be achieved by being more succinct in your discussion points and by potentially reducing some unnecessary references. The second reviewer also felt that your study would have been more compelling if you had incorporated gas-chromatography and mass spectrometry to obtain pure compounds and used that instead of just the extracts. Perhaps you can briefly address this point in your discussion section? These points aside, you have done a nice job of addressing the concerns of the previous reviews and I appreciate your efforts on your manuscript. I look forward to receiving a finalized version of this paper.

-Reply

We greatly appreciate these comments. We deleted some redundant explanations in discussion section and referred to gas-chromatography and mass spectrometry in the last paragraph. We also deleted some references as suggested.

Reviewers' Comments to Author:

Reviewer: 2

I found that the revised manuscript is substantially improved following the comments from the reviewers. I would like to request a couple of additional revisions.

-Reply

Thank you very much for providing helpful suggestions. We have corrected all the points as follows.

P9, L36 and some other places
extracted reagents -> extracts

-Reply

We corrected the wording.

P10, L26; P11, L35

alive -> live

-Reply

We corrected the word.

P12, L10

30 μ l -> ca. 30 μ l

-Reply

We corrected the wording.

P12, L11

2 individuals/ μ l -> ca. 2 individual equivalents/ μ l

-Reply

We corrected the wording.

P12, L15

4-individual-equivalent -> 4 individual equivalents

-Reply

We corrected the wording.

P12, L45

the expected ratio of male-to-female encounters is 2:1 -> the expected ratio of homosexual and heterosexual encounters is 1:2. (? Please confirm this sentence again.)

-Reply

Because two males and one female were placed in a Petri dish, we rephrased the sentence as “the expected ratio of homosexual (i.e. male meets male) and heterosexual (i.e. female meets male) encounters is 1:2”.

P15, L24

This finding implies that extracted lipid composition could make weevils stay longer on a substrate that is coated with extracted compounds from the body surface of the weevils -> This finding implies that extracts from the body surface could make weevils stay longer on a substrate that is coated with them

-Reply

We corrected the sentence as suggested.

Reviewer: 3

The text can be reduced significantly since most of the statements were overly described and can be shortened.

-Reply

We tried to concise the discussion section according to the advice.

P11, L4

Delete the sequence of...

-Reply

We deleted the words as suggested.

P11, L35

Replace with live.

-Reply

We replaced the word.

P11, L36

Did you do only one observation or several observations. If several use plural and change was to were.

-Reply

We changed the phrase as suggested because we observed the behaviour many times.

Other points:

- We deleted "...no female undertook any behaviours categorized as Stage III" since Stage III behaviour is impossible in females.
- We modified the explanation of Stage II (P11, L5-) so as to clarify the meaning.
- We corrected formats in references.